# Viral suppression among patients in HIV/AIDS care at healthcare facilities in Ethiopia: Same-day antiretroviral initiation

**Kidanu Hurisa Chachu**◉*, **Kefiloe Adolphina Maboe**

Department of Health Studies, College of Human Sciences, University of South Africa, Pretoria, South Africa

* 67120369@mylife.unisa.ac.za

## Abstract

To achieve the 2030 goal of eradicating the HIV epidemic, the Joint United Nations Programme on HIV/AIDS has set 95-95-95 targets: 95% of HIV-infected individuals should know their status, 95% should initiate antiretroviral therapy (ART), and 95% should achieve virologic suppression. In Ethiopia, progress has been made, but challenges remain. This study evaluates same-day ART initiation and its effect viral suppression for patients in HIV/AIDS care in Ethiopia. A cross-sectional study design was used to analyze 332 clinical records of patients who initiated same-day ART between October 1, 2017, and October 30, 2019. A probability simple random sampling technique was used in the selection of the clinical records of patients started on ART. Data were analyzed using Statistical Package for Social Science (SPSS) version 28.0, employing both descriptive and inferential statistical analyses. The results showed that viral suppression rates at 6, 12, and 24 months were 93%, 95%, and 86%, respectively, indicating progress towards the global target of 95% by 2030. This supports the effectiveness of same-day ART initiation in achieving viral suppression. Chi-square test results indicated no significant relationship between gender and suppression status ($p = 1.00$), age and suppression status ($p = 0.876$), or WHO stage and suppression status ($p = 0.404$). However, significant associations were found between resident type and suppression status ($p < 0.05$), as well as retention in care and suppression status ($p < 0.05$). No significant association was observed between HIV disclosure status and suppression status ($p = 0.072$). The results suggest an urgent need for follow-up and monitoring of viral suppression as months on ART increase. Additionally, enhancing viral load testing coverage at 6, 12, and 24 months is crucial to ensure sustained viral suppression.

## Introduction

The first evidence of the human immunodeficiency virus (HIV) epidemic in Ethiopia was detected in 1984 [1]. Since then, millions of Ethiopians have fallen prey to HIV/AIDS, which has also left behind hundreds of thousands of orphans. The government of Ethiopia took several steps towards preventing further disease spread and increasing accessibility to HIV prevention,

**Data availability statement:** The data that support the results of this study are available from the Unisa Institutional repository: https://uir.unisa.ac.za/handle/10500/31648. Raw data can be accessed from the corresponding author upon request.

**Funding:** The author(s) received no specific funding for this work.

**Competing interests:** The authors have declared that no competing interests exist.

care, treatment, and support for people living with HIV. The Federal Ministry of Health Ethiopia [1] reported that approximately 414,854 adults and 21,146 children below the age of 15 years took antiretroviral (ARV) drugs in Ethiopia in 2017. According to the country factsheets of the UNAIDS, the number of people receiving ART in Ethiopia was 500,000 in 2022 [2].

The consolidated guidelines of the World Health Organization (WHO) recommended the use of antiretroviral drugs for treating and preventing HIV infection and the rapid antiretroviral therapy (ART) initiation recommendation [3]. These recommendations support initiatives and measures such as early ART initiation, including same-day ART initiation regardless of WHO staging. It is against this context that Ethiopia developed consolidated national guidelines for protracted HIV prevention, care and treatment and started to implement same-day ART initiation in October 2017 [1]. According to the Federal Ministry of Health Ethiopia [1], it is critical for people living with HIV to initiate ART as early as possible, including same-day ART initiation. This will reduce the time between the diagnosis of HIV and ART initiation, thereby significantly reducing mortality and morbidity linked to HIV, as well as forward transmission of HIV, including mother-to-child transmission [1].

The HIV epidemic in Ethiopia is heterogeneous by gender, geographic area, and population group. Furthermore, HIV prevalence is seven times higher in urban areas, at 2.9% as opposed to 0.4% among both men and women in rural areas. In addition, HIV prevalence among women in urban areas is 3.6%, as opposed to 0.6% among rural women. Seven out of the nine regional states and two city administrations have HIV prevalence rates above 1%. Identification of HIV prevalence by region shows that it is highest in Gambella (4.8%), followed by Addis Ababa (3.4%), Dire Dawa (2.5%), and Harari (2.4%) [4]. The outcomes of some recent randomized trials have shown that rapid ART initiation, including same-day initiation, could improve program outcomes, especially by lessening loss to care in the pre-ART period [5]. However, evidence from program settings suggests that rapid ART initiation may result in optimized loss to follow-up after ART initiation due to insufficient time to accept and disclose HIV status and prepare for lifelong treatment [6].

Ethiopia is one of the Sub-Saharan African countries that has committed to reaching 90% of HIV-positive people, initiating ART for 90% of those reached, virally suppressing 90% of those on ART by 2020, and ending the HIV epidemic by 2030. To achieve the three 90's by 2020 and end the HIV epidemic by 2030, Ethiopia is implementing same-day ART initiation and differentiated service delivery (DSD) [1].

In Ethiopia, 79% of people living with HIV (PLHIV) were aware of their status, and 71% of eligible people living with HIV are on ART, while 87% of those on ART had attained viral suppression by May 2018. However, the viral load service coverage is 51% (Federal HIV Prevention and Control Office [4]. Ethiopia started to implement same-day ART initiation as a new initiative for all patients ready to start ART since October 2017, following WHO recommendations and developed consolidated ART guidelines as a guiding principle [1]. This study's purpose was to evaluate same-day ART initiation status regarding viral suppression in HIV/AIDS care at selected healthcare facilities in Ethiopia. The results of this study will assist healthcare providers, program managers, and federal policy designers in understanding and addressing factors associated with same-day ART initiation and viral load monitoring in HIV/AIDS care within Ethiopia's healthcare services.

## Materials and methods

### Study design and setting

A cross-sectional study design involving retrospective document analysis of 332 clinical records was used. This study was conducted in two health care facilities in Adama and Bishoftu towns in Oromia

Regional State, East Shewa zone of Ethiopia. The population for this study included the clinical records of patients who started on ART between October 01, 2017, and October 30, 2019, in two healthcare facilities in Ethiopia. The inclusion criteria encompassed the clinical records of patients who initiated ART on the same-day between October 01, 2017 and October 30, 2019. Additionally, the study included individuals belonging to adult age groups, defined as those aged 18 years and older.

## Data source and collection

A probability simple random sampling technique was used in the selection of the clinical records of patients started on ART between October 01, 2017, and October 30, 2019 from selected healthcare facilities. In sample size calculation a statistician assisted, and the Rao Soft formula was used to estimate the ideal sample size from each healthcare facility (Raosoft formula online [7]). The total sample size from both healthcare facilities was 332. The inclusion criteria encompassed the clinical records of patients who initiated ART on the same-day between October 01, 2017 and October 30, 2019. Additionally, the study included individuals belonging to adult age groups, defined as those aged 18 years and above. The data for this study were collected from the clinical records of patients who started on same-day ART from smart care databases of selected healthcare facilities. The researcher accessed clinical records from May 22, 2023, to June 05, 2023, during data collection with no access to information that could identify individual participants. To mitigate the risk of COVID-19 exposure, classified as level 1, data collection was conducted using smart care databases during this period.

## Data analysis

The collected data was entered into SPSS version 28 for data analysis. Data accuracy and quality were ensured through a thorough cleaning and preparation process conducted before data analysis. Data analysis involved a chi-square test to examine the relationships between factors associated with viral suppression for patients started on same-day ART.

In this study, validity was assured through the application of external, internal, content, and face validity to the data collection tool. To ensure reliability, several steps were taken. These encompassed assessing a data collection checklist for its clarity and alignment with research inquiries. Additionally, both the supervisor and statistician scrutinized the instrument's reliability during the final phase. Quantitative data was directly sourced by the researcher from the smart care database.

The study variables included gender, age, WHO staged at enrolment, residential status, retention in HIV care status, HIV disclosure status at enrolment, functional status and functional status. These variables were used to describe participants' health conditions, immune status, disease progression, and relevant factors that may influence viral load suppression for patients started on same-day ART.

Data was checked for errors, missing values, quality, consistency, suitability, and inconsistencies, which were identified and fixed before data analysis. The data was analyzed using descriptive and inferential statistics. Descriptive statistics were employed to summarize key characteristics of the study population, including demographics, treatment initiation status, and viral suppression. With the assistance of a statistician, the researcher used frequencies and percentages to summarize the results using tables and graphs. Data analysis involved conducting Chi-square test to examine the relationships between factors associated with viral suppression in HIV care.

## Ethical considerations

The researcher obtained ethical clearance from the University of South Africa's Department of Health Studies (reference number: HSHDC/977/2020). Following this, approval to

conduct the study was secured from the Oromia Regional Health Bureau and the participating healthcare facilities. Data collection began only after all necessary permissions were granted. As the study involved clinical records of patients who initiated same-day ART, the approval from healthcare facilities also served as consent to access these records. To ensure confidentiality and privacy, the researcher did not have access to identifiable participant information. Data clerks had assigned unique codes to each clinical record, replacing patient identification with unique numbers. Additionally, the researcher signed a confidentiality agreement before accessing the SmartCare database, reinforcing the commitment to maintaining privacy throughout the data collection process.

## Results

### Sociodemographic characteristics of participants

A total of 332 clinical records of patients aged over 18 who started on same-day ART were analyzed in this study. The majority of participants were female (52%, n = 172), and 48.5% (n = 161) were married. Additionally, 58.7% (n = 195) were over 35 years old, and 29.5% (n = 98) were in the age range of 25–34 years. The study also revealed that 61.7% (n = 205) of the participants were Orthodox Christians, 38.3% (n = 127) had primary education, 72% (n = 239) were from urban areas, and 88.6% (n = 294) had their phone numbers documented (Table 1).

### Baseline clinical and laboratory information

The variables included in the baseline clinical and laboratory information were patients' histories of opportunistic illness, types of opportunistic infections, baseline BMI, functional status, WHO clinical staging of HIV, patients' HIV disclosure status, patients' CD4 cell count at baseline, and the actual CD4 value at baseline. The majority, 94.3% (n = 313) patients, had no opportunistic infections, and 68% (n = 13) of opportunistic infections were tuberculosis.

The results indicated that the majority of participants, 63.3% (n = 210), had a normal nutritional status, while 7.5% (n = 25) were overweight, 6.6% (n = 22) had moderate malnutrition, 6% (n = 20) had severe malnutrition, and 2.4% (n = 8) were obese. Additionally, 90.7% (n = 301) had a normal functional status (Fig 1). Regarding the WHO staging, 54% (n = 180) were in stage I, 26% (n = 87) were in stage II, 15% (n = 51) were in stage III, and 4% (n = 14) were in stage IV.

Our results indicated that the majority, 53.6% (n = 178), of patients did not inform anyone about their HIV status, including their spouses or families (Fig 2). Regarding baseline CD4 results, the majority, 65.1% (n = 216), did not have baseline CD4 results, while 34.9% (n = 116) did. Among those with CD4 results, 29.3% (n = 34) had CD4 counts below 200 cells/mm³, 26.7% (n = 31) had counts between 200-349 cells/mm³, 17.2% (n = 20) had counts between 350–499 cells/mm³, and 26.7% (n = 31) had counts of 500 cells/mm³ or higher.

### Same-day ART initiation status related information

The HIV testing unit plays a crucial role in facilitating same-day initiation of ART. This includes patients identified as HIV positive in other healthcare facilities and referred for ART initiation, which can sometimes cause delays. Fig 3 shows that the majority of patients, 27.7% (n = 92), were from voluntary counselling and testing (VCT), and 19.9% (n = 66) were referred from other healthcare facilities.

Fig 4 shows that the majority of patients, 48% (n = 160), were in HIV care for less than 6 months, indicating that a significant number of those who started on same-day ART were lost to follow-up before reaching the six-month period.

**Table 1. Sociodemographic characteristics of participants at baseline, by study variables (N = 332).**

| Variables | | Frequency (N) | Percentage (%) |
|---|---|---|---|
| Age | 18–24 years | 39 | 11.7 |
| | 25–34 years | 98 | 29.5 |
| | >35 years | 195 | 58.7 |
| Sex | Male | 160 | 48.2 |
| | Female | 172 | 51.8 |
| Marital status | Single | 71 | 21.4 |
| | Married | 161 | 48.5 |
| | Divorced | 72 | 21.7 |
| | Widowed | 26 | 7.8 |
| | Separated | 2 | 0.6 |
| Religion | Protestant | 61 | 18.4 |
| | Catholic | 22 | 6.6 |
| | Orthodox | 205 | 61.7 |
| | Muslim | 38 | 11.4 |
| | Others specify (Wakefeta) | 6 | 1.8 |
| Educational level | No formal education | 95 | 28.6 |
| | Primary | 127 | 38.3 |
| | Secondary | 83 | 25.0 |
| | Tertiary | 27 | 8.1 |
| Patients address | Urban | 239 | 72 |
| | Rural | 93 | 28 |
| House number documented | Yes | 71 | 21 |
| | No | 261 | 79 |
| Patient have phone number | Yes | 294 | 89 |
| | No | 38 | 11 |

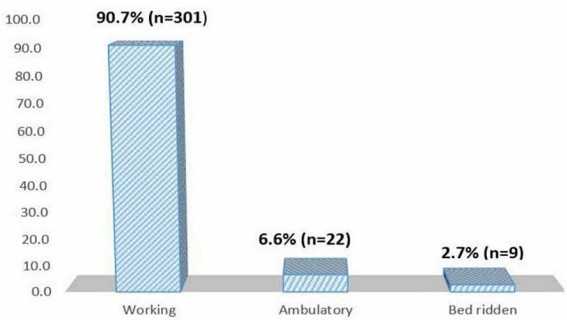

**Fig 1. Baseline functional status of participants (N = 332).**

The results indicated that the majority, 59% (n = 196), remained in HIV care, while 27% (n = 90) were lost to follow-up, 7% (n = 25) were confirmed dead, and 6% (n = 21) were transferred to another healthcare facility (Fig 5).

## Viral load test and suppression related information

Our results showed that the majority of patients were not eligible for a viral load test: 41.3% (n = 137) at 6 months, 61.4% (n = 204) at 12 months, and 96.7% (n = 321) at 24 months (Table 2).

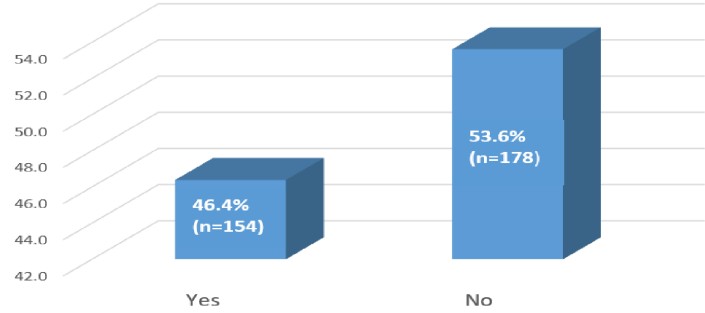

**Fig 2. HIV disclosure status of patients at enrolment to same-day ART initiation (N = 332).**

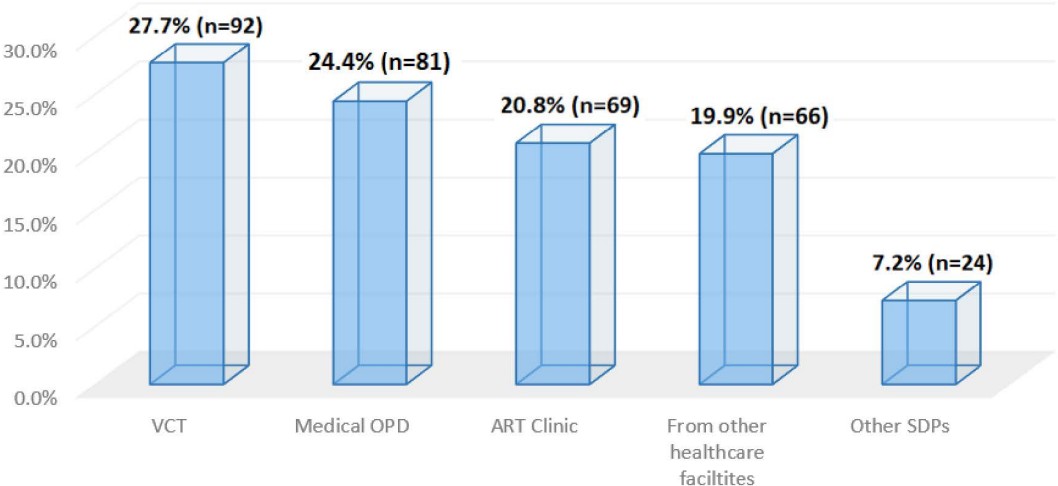

**Fig 3. HIV testing and diagnosis unit of patients started on same-day ART (N = 332).**

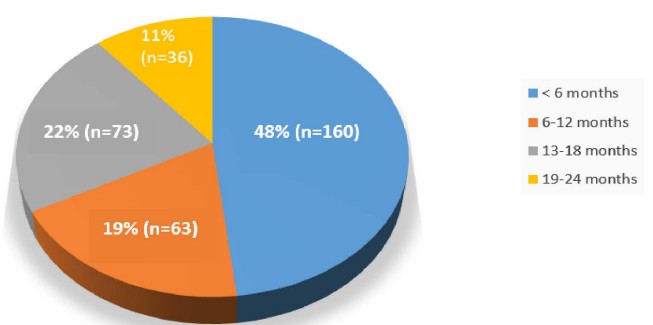

**Fig 4. Months on ART since ART started (N = 332).**

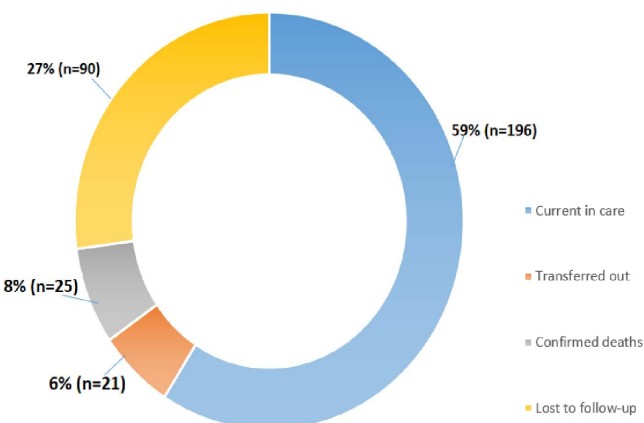

**Fig 5. HIV care and retention status of patients (N = 332).**

The viral load suppression cut point of less than 1000 copies/ml was used for viral load suppression based on the Ethiopian National Consolidated Guidelines for comprehensive HIV prevention, care and treatment [1]. Fig 6 showed that the majority, 93% (n = 126), of patients have suppressed viral load test results, while the minority, 7% (n = 9) have high viral load results. These results implied that viral load suppression was promising progress towards achieving the target of 95% set for 2030.

Fig 7 below shows viral road test results at 12 months among patients who had the test performed at 12 months. The majority 95% (n = 77) of patients have suppressed viral load results.

Fig 8 below provides an overview of the viral load test results at the 24-month mark for a group of patients whose viral load test was done. The majority of 86% (n = 6) patients had achieved suppressed viral load results.

The chi-square test results indicated no significant relationship between gender and suppression status (p = 1.00), while a significant association was found between resident type and suppression status (p < 0.05). The results also showed no significant association between HIV disclosure status and suppression status (p = 0.072), aligning with findings from Ethiopia (Table 3).

## Discussion

This study assesses the status of same-day ART initiation concerning viral suppression for patients in HIV/AIDS care at healthcare facilities in Ethiopia. The results revealed that females had a higher prevalence of HIV infection compared to males, indicating their increased vulnerability. A comparable study conducted in South Africa on same-day ART initiation for HIV-infected individuals reported that female enrollment was 74.1% (n = 9663), while male enrollment was 25.9% (n = 3375) [8]. In contrast to this study, a study conducted in Addis Ababa, Ethiopia, on the quality of antiretroviral therapy services and associated factors at public hospitals found that a higher proportion of male patients, 67.1% (n = 282), were enrolled compared to female patients, who accounted for 32.9% (n = 138) [9].

Regarding marital status, the study revealed that the majority of patients were married, indicating a high prevalence of HIV among married couples due to a lack of HIV testing before and during marriage. A supporting study conducted in Ethiopia on pre-marital HIV testing among married women found that only 21.4% (n = 2142) of married couples had undergone pre-marital HIV testing [10]. In contrast to this study, a study conducted in

**Table 2. Frequency distribution viral load test status at 6, 12 and 24 months (N = 332).**

| Variables | | Frequency (N) | Percentage (%) |
|---|---|---|---|
| Viral load test status at 6 months | Yes | 135 | 40.7 |
| | No | 60 | 18.1 |
| | Not applicable | 137 | 41.3 |
| Viral load test status at 12 months | Yes | 81 | 24.4 |
| | No | 47 | 14.2 |
| | Not applicable | 204 | 61.4 |
| Viral load test status at 24 months | Yes | 7 | 2.1 |
| | No | 4 | 1.2 |
| | Not applicable | 321 | 96.7 |

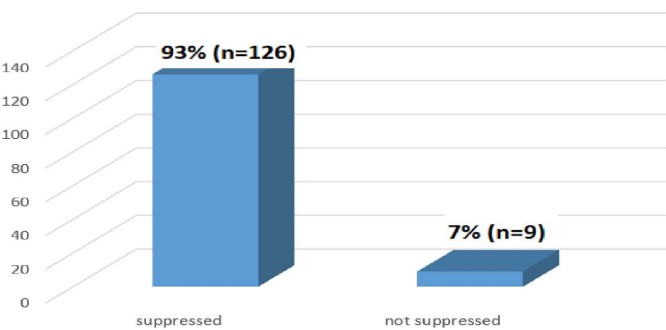

**Fig 6. Viral load test results at 6 months (N = 135).**

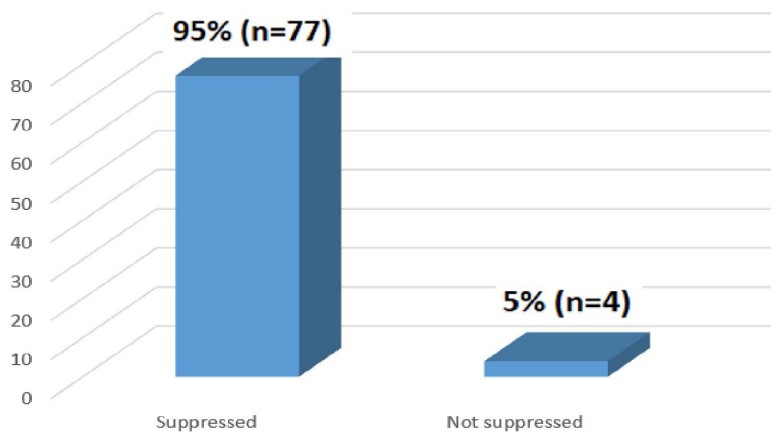

**Fig 7. Viral load test results at 12 months (N = 81).**

Eswatini on the impact of same-day ART initiation under the World Health Organization's treat-all policy found that the majority of patients on same-day ART, 69.2% (n = 566), were not married [11].

The study revealed that most patients had only a primary education, indicating a high prevalence of HIV among those with no formal or only primary education. A similar study conducted in Masaka, Uganda, on factors related to loss to follow-up (LTFU) among HIV-positive patients receiving ART found that patients with no formal education had

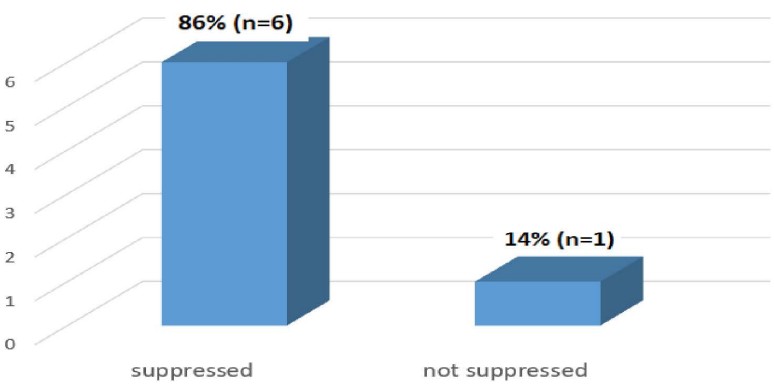

**Fig 8. Viral load test results at 24 months (N = 7).**

**Table 3. Frequency distribution of viral suppression Chi-square analysis results (N = 332).**

| Independent variables | | Suppressed | | Not suppressed | | Total | P value |
|---|---|---|---|---|---|---|---|
| | | Frequency (N) | Percentage (%) | Frequency (N) | Percentage (%) | | |
| Gender | Male | 48 | 94.1 | 3 | 5.9 | 51 | 1.00 |
| | Female | 78 | 92.9 | 6 | 7.1 | 84 | |
| Age | 18–24 | 12 | 100 | 0 | 0 | 12 | 0.876 |
| | 25–34 | 33 | 94.3 | 2 | 5.7 | 35 | |
| | 35 and above | 81 | 92 | 7 | 8 | 88 | |
| WHO stage | Stage I | 68 | 90.7 | 7 | 9.3 | 75 | 0.404 |
| | Stage II | 34 | 94.4 | 2 | 5.6 | 36 | |
| | Stage III | 23 | 100 | 0 | 0 | 23 | |
| | Stage IV | 1 | 100 | 0 | 0 | 1 | |
| Residential type | Urban | 101 | 96.2 | 4 | 3.8 | 105 | 0.026 |
| | Rural | 25 | 83.3 | 5 | 16.7 | 30 | |
| Retention status | Not retained | 4 | 57.1 | 3 | 42.9 | 7 | 0.006 |
| | Retained | 122 | 95.3 | 6 | 4.7 | 128 | |
| Functional status | Working | 117 | 92.9 | 9 | 7.1 | 126 | 1.00 |
| | Ambulatory | 8 | 100 | 0 | 0 | 8 | |
| | Bed ridden | 1 | 100 | 0 | 0 | 1 | |
| Disclosure status | Yes | 83 | 92.9 | 3 | 7.1 | 86 | 0.072 |
| | No | 43 | 100 | 6 | 0 | 49 | |

a higher risk of being lost to follow-up compared to those with post-secondary education (AHR = 0.50; 95% CI, 0.34–0.75) [11]. In contrast, a study conducted in Togo on health-related quality of life among people living with HIV/AIDS found that the majority of patients, 45.4% (n = 399), had secondary or higher education, while 37% (n = 326) had primary education, and 17.6% (n = 155) had no education. This indicates that, unlike the findings of this study, a higher proportion of patients in the Togo study had secondary or higher education [12].

The study found that most patients had their phone numbers documented. This suggests that the majority of patients could be contacted if they were lost through their phone numbers, with female patients being more likely to have a phone number and be reachable compared to male patients. A similar study conducted in Kampala, Uganda, on factors associated

with retention and non-viral suppression among HIV-positive patients on ART found that 94.2% (n = 259) of patients had a phone number, while only 5.8% (n = 16) did not [13].

The study revealed that most patients had a registered house number. However, these results suggest that many patients did not have their exact location or house number documented, which affects home-to-home tracing for those lost from HIV care. Having a specific address is crucial for effective HIV prevention, care, and treatment. A similar study conducted in rural Mozambique on loss to follow-up and opportunities for reengagement in HIV care found that 61.6% (n = 691) of patients were reported lost to follow-up due to a lack of properly documented addresses [14].

The study discovered that tuberculosis was the most common opportunistic infection among patients who started same-day ART. This meant that patients were more likely to have other illnesses and not follow their treatment plan because they had so many pills to take. A related study conducted in Haiti on the importance of integrated care for HIV and TB co-infection revealed that prompt initiation of both ART and TB treatment can improve outcomes and address the dual burden of HIV and TB. It was found that 77.1% (n = 37) of patients who started ART on same-day ART had a TB infection at enrollment [15]. In contrast, a study conducted at Gondar University Comprehensive and Specialized Hospital in Ethiopia on the incidence of opportunistic infections and their predictors among HIV/AIDS patients found that the majority of patients had pneumocystis pneumonia (16.51%, n = 90), chronic diarrhea (16.33%, n = 89), bacterial pneumonia (10.82%, n = 59), and pulmonary tuberculosis (10.46%, n = 57). This indicates that tuberculosis was the fourth most common opportunistic infection [16]. Another study that differed from this one, conducted in Kinshasa, Democratic Republic of Congo, found that the majority of patients had malaria (45.4%, n = 54), while 29.4% (n = 35) had tuberculosis. This suggests that tuberculosis was the second most common opportunistic infection in that study [17].

The study indicated that most patients were in WHO stage I, showing that a significant portion of those who began same-day ART had early-stage HIV infections with minimal symptoms. This early stage allowed for easier medication adherence and reduced the potential for stigma and discrimination. In contrast, patients who were ambulatory or bedridden required specialized care and additional resources to ensure medication adherence. A similar study conducted at Nekemte Specialized Hospital in Western Ethiopia, which focused on same-day ART initiation and its associated factors, found that 77.02% (n = 372) of patients were in stage I, 14.08% (n = 67) were in stage II, and 8.91% (n = 43) were in stages III and IV [18]. In this regard, the results of this study differ from those of a study conducted in Malang, East Java, Indonesia, on functional status and the incidence of loss to follow-up after ART initiation. That study found that the majority of patients were in WHO stage III (35.8%, n = 53) or WHO stage IV (27%, n = 40), with 22.9% (n = 34) in WHO stage I and 14.1% (n = 21) in WHO stage II. This indicates that the majority of patients in the Malang study were in WHO stage III, whereas in this study, the majority were in WHO stage I [19].

The study revealed that the majority of patients had been in HIV care for less than 6 months, indicating that a significant number of those who started same-day ART were lost to follow-up before reaching the six-month mark. A similar study conducted in South African public health facilities on same-day ART initiation reported that 33% (n = 11,114) of patients were classified as lost to follow-up, with a median time to loss of 55 days [20]. According to Joseph Davey et al [20], the results showed a retention rate of approximately 67% (n = 22,565) at six months, which aligns with the results of this study.

The study indicated that the majority of patients were lost to follow-up, suggesting that Ethiopia is significantly below the UNAIDS target of achieving 95% retention in HIV care by 2030. The findings also highlighted the need for increased efforts to retain patients in

HIV care, especially in the first six months after same-day ART initiation. A similar study conducted in South Africa on same-day ART initiation for HIV-infected adults found that 64.4% (n = 8399) of patients remained active in care, 29.2% (n = 3804) were lost to follow-up, 6.1% (n = 793) were transferred to other healthcare facilities, and 0.3% (n = 42) had died [8]. Similarly, data from 72 countries as part of a global update on progress towards the 90–90–90 targets for ending AIDS revealed that retention on antiretroviral therapy after 12 months ranged from 72% in Western and Central Africa to 89% in the Middle East and North Africa. These findings are consistent with the results of this study [21]. However, a study conducted at the Kibera Community Health Center HIV/AIDS program in Kenya on patient retention in HIV care found that the majority of patients, 79% (n = 67), were still in care. Additionally, 14% (n = 12) were lost to follow-up, 6% (n = 5) were transferred to other healthcare facilities, and 1% (n = 1) had died [22].

The study revealed that only about 40% (n = 135) of patients had a viral load test conducted 6 months after same-day ART initiation. These results indicate low coverage of viral load testing at 6 months, highlighting the need for healthcare providers to focus more on ensuring viral load tests are performed. A similar study conducted in Ekurhuleni District, South Africa, found that 64.1% (n = 455) of patients initiated on same-day ART had their viral load tested at 6 months, showing a comparable result to this study [23]. In Johannesburg, South Africa, a study on clinical predictor scores for identifying patients at risk of poor viral load suppression at six months on ART found that 80.7% (n = 239) of patients had a viral load test at six months. This percentage was higher compared to the viral load testing coverage observed in this study [24].

The results showed that 24.4% (n = 81) of patients had documented viral load results at 12 months. This indicates that many patients missed their viral load tests at this critical time point, which is crucial for monitoring treatment progress. In contrast, a study conducted in Myanmar on the performance and outcomes of routine viral load testing in people living with HIV found that the majority, 66.1% (n = 4731), of patients had viral load test results, reflecting a higher coverage compared to this study [25].

The study found that only 2.1% (n = 7) of patients had documented viral load results at 24 months. This low coverage suggests a need for healthcare providers to focus on improving patient retention in care to reduce morbidity and mortality. A similar study conducted in the Hlabisa sub-district of South Africa on clinical outcomes after first-line HIV treatment revealed that only 25.5% (n = 4334) of patients had viral load results at 24 months [26]. A study conducted in Malawi on viral load monitoring outcomes from a decentralized HIV program found that 89% (n = 10,476) of patients had viral load results at 24 months. This was significantly higher compared to the results of this study [27].

A viral load suppression threshold of less than 1,000 copies/ml was used for evaluating viral load suppression, according to the Ethiopian National Consolidated Guidelines for Comprehensive HIV Prevention, Care, and Treatment [1]. The study results showed that the majority, 93% (n = 126), of patients have suppressed viral load test results. Similarly, a study conducted in Ekurhuleni District, South Africa, on the feasibility of implementing same-day ART initiation during routine care, concurred with this study and indicated that the viral suppression rate at 6 months was 78.1% (n = 118) [23]. A study conducted in rural Lesotho compared ART refills provided by community health workers versus clinic-based follow-up after home-based same-day ART initiation. It found lower rates of viral suppression, with only 44% (n = 112) achieving viral suppression at six months. This result contrasts with the findings of this study [28].

The study found that the majority, 95% (n = 77), of patients had suppressed viral load results at 12 months. A similar study conducted in Lilongwe, Malawi, on outcomes for

women newly initiated on lifelong antiretroviral therapy during pregnancy, reported that 90% (n = 269) of patients achieved viral suppression at 12 months [29]. The results of this study differ from those of a study conducted in rural Lesotho, which compared ART refills provided by community health workers versus clinic-based follow-up after home-based same-day ART initiation. That study reported a 54% (n = 138) viral suppression rate at 12 months, which was lower compared to the viral suppression rate observed in this study at the same time point [28].

The study revealed that 86% (n = 6) of patients had achieved suppressed viral load results at 24 months. These results suggest promising progress toward meeting the UNAIDS 2030 target. A similar study conducted in Vietnam, which assessed HIV viral load monitoring in remote settings, found that 93.4% (n = 340) of patients achieved viral suppression at 24 months [30]. Another study that supported these findings was conducted in Lilongwe, Malawi, on outcomes for women newly initiated on lifelong antiretroviral therapy during pregnancy. This study revealed that at 24 months, the majority, 91% (n = 271), of patients achieved viral suppression [29].

## Limitation of the study

The study faced several limitations exacerbated by COVID-19 pandemic restrictions, notably hindering face-to-face data collection. Additionally, its retrospective nature introduces potential recall bias among patients recalling details of same-day ART initiation and follow-up events. The cross-sectional design limited tracking longitudinal changes in patient outcomes. Furthermore, restrictions during the pandemic prevented inclusion of religious leaders and community-level associations of people living with HIV. Recognizing these limitations is crucial for informing future research and refining understanding of same-day ART initiation and associated challenges.

## Conclusion

This study evaluated the impact of same-day ART initiation on viral suppression in Ethiopian HIV/AIDS care. Viral suppression rates at 6, 12, and 24 months were 93%, 95%, and 86%, respectively, indicating progress towards the global target of 95% by 2030. These results support the effectiveness of same-day ART initiation in achieving viral suppression. Implementing same-day ART initiation strategies can improve patient outcomes in Ethiopian HIV care regarding viral suppression. The study also found that viral load testing rates at 6, 12, and 24 months were low, affecting viral suppression. Addressing these challenges through targeted interventions can enhance HIV care and support the goal of eradicating the epidemic by 2030. The study emphasizes a holistic approach to same-day ART initiation, recommending improved counseling, capacity building, and education to address patient concerns and boost adherence.

## Acknowledgments

We extend our heartfelt gratitude to Ms. Princess Lekhondlo Masondo, a statistician at UNISA, for her invaluable assistance with the data analysis. Her expertise was crucial to the success of this research. We are also deeply thankful to the University of South Africa, the Oromia Regional Health Bureau, and the selected healthcare facilities for granting approval for this study. Our appreciation further goes to the data clerks and the human resources unit at the healthcare facilities for their steadfast support in facilitating data collection.

## Author contributions

**Conceptualization:** Kidanu Hurisa Chachu.

**Data curation:** Kidanu Hurisa Chachu.

**Formal analysis:** Kidanu Hurisa Chachu.

**Investigation:** Kidanu Hurisa Chachu.

**Methodology:** Kidanu Hurisa Chachu, Kefiloe Adolphina Maboe.

**Project administration:** Kidanu Hurisa Chachu.

**Resources:** Kidanu Hurisa Chachu.

**Software:** Kidanu Hurisa Chachu.

**Supervision:** Kefiloe Adolphina Maboe.

**Validation:** Kidanu Hurisa Chachu, Kefiloe Adolphina Maboe.

**Visualization:** Kidanu Hurisa Chachu.

**Writing – original draft:** Kidanu Hurisa Chachu.

**Writing – review & editing:** Kidanu Hurisa Chachu, Kefiloe Adolphina Maboe.

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
