## [Decision Letter · Decision Letter 0]

4 Nov 2024

PONE-D-24-34271Viral suppression among patients in HIV/AIDS care at healthcare facilities in Ethiopia: Same-day antiretroviral initiationPLOS ONE

Dear Dr. Chachu,

Thank you for submitting your manuscript to PLOS ONE. After careful consideration, we feel that it has merit but does not fully meet PLOS ONE’s publication criteria as it currently stands. Therefore, we invite you to submit a revised version of the manuscript that addresses the points raised during the review process.

**Thank you for the submission. After careful review, we find it has merit.** Please submit your revised manuscript by Dec 19 2024 11:59PM. If you will need more time than this to complete your revisions, please reply to this message or contact the journal office at plosone@plos.org . Please include the following items when submitting your revised manuscript:

We look forward to receiving your revised manuscript.

Kind regards,

Judith Kose, M.D.

Academic Editor

PLOS ONE

Journal Requirements:

3. In the online submission form, you indicated that [Insert text from online submission form here]. 

4. Please remove your figures from within your manuscript file, leaving only the individual TIFF/EPS image files, uploaded separately. These will be automatically included in the reviewers’ PDF.

Additional Editor Comments:

Thank you for your submission. Please see below the reviewer comments from the reviewers for you to address.

1st Reviewer:

a) The authors describe VL<1000c/ml as VLS, WHO says this level results in practically zero transmission and HIV <200 as undetectable and <50/20/5 copies as viral suppression. This is not possible with this set of data as like SA- VL is defined as suppression if <1000c/ml.

It is thus not current to call HIV loads<1000 as Viral suppression.- Thus the authors need to clarify this.

b)

The result and conclusion sectors needs to be re written to make it more concise.

c) Only 7 patients were followed up to 24months- and thus too small a sample. The massive drop out rate is not discussed either

d) The number of people infected and on therapy is not clear.

e) For a country to know what it needs to achieve to reach the WHO 95-95-95 targets for epidemic control- it needs to have data on numbers infected, numbers initiated, numbers remaining in care as well as suppression rates or numbers on treatment who have rates that will impact on transmission. Critical would be to recognise what is needed to address the challenges of retention- added counselling, intensive adherence support, community based support etc

2nd reviewer

a) The manuscript is technically sound, methodology is clear, and data is presented well. The findings follow logically from the hypothesis to the conclusion. The data collected is in line with the objectives of the study and supports the authors’ claims.

Statistical analysis in the manuscript is proper and thorough. Authors have used suitable statistical methods which are clearly described, and results are reproducible and valid. They have also addressed the confounding variables which makes the findings more reliable.

Authors have made all data fully available following best practices in transparency in research. They have included a data availability statement which describes where and how the data can be accessed so that others can reuse the data and verify their findings.

The manuscript is well organized and written in clear language. It follows standard English and is accessible to a wide audience. Section flow is good and makes complex concepts easy to understand

The following needs revision:

The reference list requires attention, specifically reference number 3, which cites the World Health Organization (WHO). It is important to ensure that the name is formatted correctly and consistently throughout the manuscript. Please revise this entry to accurately reflect the proper citation format for WHO.

Reviewers' comments:

Reviewer's Responses to Questions

**Comments to the Author**

1. Is the manuscript technically sound, and do the data support the conclusions?

Reviewer #1: Partly

Reviewer #2: Yes

2. Has the statistical analysis been performed appropriately and rigorously? 

Reviewer #1: No

Reviewer #2: Yes

3. Have the authors made all data underlying the findings in their manuscript fully available?

Reviewer #1: Yes

Reviewer #2: Yes

4. Is the manuscript presented in an intelligible fashion and written in standard English?

Reviewer #1: Yes

Reviewer #2: Yes

5. Review Comments to the Author

Reviewer #1: The challenge facing authors in Africa is the changing goal posts especially in regard to what defines HIV load suppression:

The authors describe VL<1000c/ml as VLS, WHO says this level results in practically zero transmission and HIV <200 as undetectable and <50/20/5 copies as viral suppression. This is not possible with this set of data as like SA- VL is defined as suppression if <1000c/ml.

It is thus not current to call HIV loads<1000 as Viral suppression.- Thus the authors need to clarify this.

The result and conclusion sectors need a major rewrite.

Only 7 patients were followed up to 24months- and thus too small a sample.

The massive drop out rate is not discussed either

the number of people infected and on therapy is not clear.

For a country to know what it needs to achieve to reach the WHO 95-95-95 targets for epidemic control- it needs to have data on numbers infected, numbers initiated, numbers remaining in care as well as suppression rates or numbers on treatment who have rates that will impact on transmission. Critical would be to recognise what is needed to address the challenges of retention- added counselling, intensive adherence support, community based support etc

Reviewer #2: The manuscript is technically sound, methodology is clear, and data is presented well. The findings follow logically from the hypothesis to the conclusion. The data collected is in line with the objectives of the study and supports the authors’ claims.

Statistical analysis in the manuscript is proper and thorough. Authors have used suitable statistical methods which are clearly described, and results are reproducible and valid. They have also addressed the confounding variables which makes the findings more reliable.

Authors have made all data fully available following best practices in transparency in research. They have included a data availability statement which describes where and how the data can be accessed so that others can reuse the data and verify their findings.

The manuscript is well organized and written in clear language. It follows standard English and is accessible to a wide audience. Section flow is good and makes complex concepts easy to understand

The following needs revision:

The reference list requires attention, specifically reference number 3, which cites the World Health Organization (WHO). It is important to ensure that the name is formatted correctly and consistently throughout the manuscript. Please revise this entry to accurately reflect the proper citation format for WHO.

6. PLOS authors have the option to publish the peer review history of their article (what does this mean? ). If published, this will include your full peer review and any attached files.

**Do you want your identity to be public for this peer review?** For information about this choice, including consent withdrawal, please see our Privacy Policy .

Reviewer #1: No

Reviewer #2: **Yes: ** Dr. Alemayehu Abebe Demissie

---

## [Author Response · Author response to Decision Letter 1]

27 Nov 2024

The detailed responses to the reviewers' comments have been provided as an attachment. We would like to extend our sincere thanks to both the reviewers and the editors for their valuable feedback and support.

---

## [Editor Report · Decision Letter 1]

13 Dec 2024

Viral suppression among patients in HIV/AIDS care at healthcare facilities in Ethiopia: Same-day antiretroviral initiation

PONE-D-24-34271R1

Dear Dr. Chachu,

We’re pleased to inform you that your manuscript has been judged scientifically suitable for publication and will be formally accepted for publication once it meets all outstanding technical requirements.

Kind regards,

Judith Kose, M.D.

Academic Editor

PLOS ONE
---

## [Editor Report · Acceptance letter]

PONE-D-24-34271R1

PLOS ONE

Dear Dr. Chachu,

I'm pleased to inform you that your manuscript has been deemed suitable for publication in PLOS ONE. Congratulations! Your manuscript is now being handed over to our production team.

Kind regards,

on behalf of

Dr. Judith Kose

Academic Editor

PLOS ONE
